# Impact of Surgical Timing on Outcomes in Neonatal Inguinal Hernia Repairs: A Systematic Review

**DOI:** 10.3390/pediatric17010012

**Published:** 2025-01-23

**Authors:** Leen Yahya Alqahtany, Arwa Alsharif, Abdulaziz Alsharif, Omar Alanazi, Manaf Altaf, Ahlam Kaleemullah, Lana Alsaedi, Hanan Ismail Wasaya, Abrar Hassan Alharbi, Osama Bawazir

**Affiliations:** 1Department of Medicine and Surgery, Batterjee Medical College, Jeddah 21442, Saudi Arabia; 150024.leen@bmc.edu.sa (L.Y.A.); 140300.ahlam@bmc.edu.sa (A.K.); 140310.lana@bmc.edu.sa (L.A.); 140204.hanan@bmc.edu.sa (H.I.W.); 140023.abrar@bmc.edu.sa (A.H.A.); 2Department of Medicine and Surgery, Vision College, Jeddah 23643, Saudi Arabia; 202313034@vision.edu.sa (A.A.); 202313068@vision.edu.sa (M.A.); 3College of Medicine, King Saud Bin Abdulaziz University for Health Sciences, Riyadh 11426, Saudi Arabia; alanazi323@ksau-hs.edu.sa; 4Department of Surgery, King Faisal Specialist Hospital and Research Center, Jeddah 12713, Saudi Arabia; obawazeer@kfshrc.edu.sa

**Keywords:** inguinal hernia, surgical timing, neonatal surgery, complications, outcomes, pediatric surgery

## Abstract

Inguinal hernia repair (IHR) is a common surgical procedure among neonates and infants; the time of surgery is one of the major factors affecting its outcomes. Our systematic review aims to evaluate the effects of surgical timing on outcomes in inguinal hernia repairs in the newborn and infant population to establish evidence-based guidelines for optimal surgical timing. A systematic search was performed in PubMed, MEDLINE, and Web of Science databases, following PRISMA guidelines. Studies evaluating neonates and infants undergoing IHR with outcomes of recurrence, complications, and postoperative recovery were included. Data were collaboratively extracted, including patient demographics, surgical approaches, perioperative complications, and long-term outcomes. Early repair (0–28 days of life) decreased the risk of hernia incarceration but also increased the risk of preoperative complications. Delayed repair (29 days to 1 year of life) showed fewer preoperative complications but increased the risk of incarceration. The outcomes were affected by variables including patient maturity and comorbidities, along with hernia severity. Neonates with a high risk for incarceration are best treated with early repair, while stable infants can be managed safely with delayed repair. More randomized trials are needed to develop standardized guidelines that balance the associated risks of neonatal versus infant repair strategies to maximize benefits.

## 1. Introduction

Neonatal inguinal hernia is a common pathology in the neonate, occurring in approximately 1–5% of all live births, with high prevalence among preterm neonates (less than 37 weeks of gestational age) [1,2]. It is defined as the protrusion of a portion of the intestine through the inguinal canal due to incomplete closure of the vaginal process during embryological development [3]. The condition is typically treated during the neonatal period (the first 28 days of life or up to 44 weeks of gestational age) in premature neonates, or during the infant period (from 29 days up to 1 year of life) [4]. There is a risk of hernia incarceration with delayed surgical treatment (after 29 days) for a neonatal inguinal hernia, which could result in intestinal obstruction or strangulation and other long-term complications [5]. Clinical practice guidelines recommend immediate hernia repair surgery for all cases of neonatal inguinal hernia to prevent complications [6].

However, the timing of inguinal hernia repair is not specified due to the fact that the balance between lowering the risks associated with early repair (during the neonatal period) can significantly decrease the risk of incarceration, but it is still associated with an increased risk of postoperative respiratory complications and the need for prolonged hospital observation, especially in premature neonates. Previous research suggests delaying surgical repair (during the infant period) in preterm or hemodynamically unstable neonates to promote growth and stabilize the clinical condition, potentially reducing perioperative complications [7]. They have also identified several factors, including gestational age at birth, comorbidities, and timing of surgical intervention, that affect surgical outcomes regarding neonatal inguinal hernia repair [8]. However, the relative risks and benefits of surgical timing are more complex and are not necessarily age- or maturity-specific due to the ongoing physiologic development of neonates. As a result, the timing of hernia repair plays a significant role in surgical outcomes and overall success.

We conducted a systematic review with the objective to assess the effect of timing on the risk of complications and on outcomes after neonatal inguinal hernia repair and to compare early operative intervention against delayed operative intervention. However, while many studies have investigated outcomes after neonatal hernia repairs, there is still ongoing debate regarding optimal timing. Supporters of early repair point out that it lowers the risk of complications, while advocates for delayed repair argue that care of the neonate, allowing them to grow and become stable before surgery, is beneficial. This study aims to provide essential evidence-based insights that will guide the timing of surgery to maximize benefits while minimizing harm.

## 2. Materials and Methods

### 2.1. Search Strategy

The systematic review was registered in PROSPERO (CRD42024610560) and was conducted following the Preferred Reporting Items for Systematic Reviews and Meta-Analyses (PRISMA) guidelines (Appendix A) [9]. We conducted a comprehensive electronic search using the following databases, PubMed, MEDLINE, and Web of Science, without any specific time frame. A search strategy was developed by the authors O.B. and A.A., and approved by the rest of the research team. Based on the timing of surgical intervention, we utilized two groups in this review: “early repair”, which we defined as a surgical intervention within the neonatal period, which is the first 28 days of life (or up to 44 weeks gestational age for premature neonates), and “Delayed repair”, which included surgical intervention within the infant period, which is 29 days to 1 year of life. Repairs occurring after age 1 year were excluded. Studies related to the impact of surgical timing on outcomes in neonatal inguinal hernia repairs were identified inclusively using a combination of Medical Subject Headings (MeSH) such as “Neonatal Inguinal Hernia” OR “Neonatal Hernia Repair” OR “Pediatric Hernia” OR “Inguinal Hernia Surgery” AND “Surgical Timing” OR “Neonatal Intervention” OR “Infant Intervention” OR “Surgical Outcomes” OR “Complications” AND “Incarceration” OR “Postoperative Complications”. To identify any missing articles, a further review of the references for the studies was conducted.

The search technique included searching several databases: PubMed (*n* = 2167), MEDLINE (*n* = 1698), and Web of Science (*n* = 3265). At first, the records were checked for duplicates, leaving 7130 distinct records. During the eligibility phase, 1267 records were reviewed, and 5863 records were eliminated based on the established criteria. Out of the records reviewed, 305 full-text articles were evaluated for eligibility, excluding 962 articles with indicated reasons. Seventy-seven papers met the criteria for inclusion in the qualitative synthesis.

### 2.2. Study Selection

#### 2.2.1. Inclusion Criteria

This systematic review included studies that assessed the optimal timing of surgery and its effect on the outcome of neonatal inguinal hernia repairs. It included studies on early repairs (during neonatal period, which is 0–28 days of life, or up to 44 weeks gestational age for premature neonates) or delayed repairs (during infant period, which is 29 days to 1 year of life). This review included randomized controlled trials (RCTs), quasi-experimental studies, cohort studies, case–control studies, and observational studies published in English. Furthermore, this review included studies published in peer-reviewed journals or other credible sources.

#### 2.2.2. Exclusion Criteria

This systematic review excluded studies that do not assess the optimal timing of surgery and its effect on the outcome of neonatal inguinal hernia repairs. It excluded studies that included children of 1 year old and older. We also disqualified studies investigating surgical management unrelated to inguinal hernia repairs, animal investigations, and in vitro assays. This review does not include studies published in non-English languages or those with insufficient data, such as bidding outcome measures, specific numerical results of scheduling variances, or relevant statistical analysis features essential for determining the impact of surgical timing.

#### 2.2.3. Screening and Data Extraction

All records were securely stored and organized for systematic review purposes, with access restricted to the research team. Four authors (L.Y.A., A.A., O.A., and M.A.) then imported the remaining results into Rayyan [(https://www.rayyan.ai/), accessed 8 November 2024] for title and abstract screening based on relevance. Then, three authors (A.K., L.S.A., and H.I.W.) performed a full-text review of the papers that passed the first screening for inclusion or exclusion [10]. Discussions with O.B. and other researchers resolved any disagreements during the screening process. An Excel sheet was prepared by A.A., A.H.A., and O.B. to extract the following data from the selected studies, including title, author name, country, year of publication, journal name, study design, level of evidence, sample size, surgical complications, recurrence, and mortality rates.

#### 2.2.4. Quality Assessment and Bias Evaluation

The Grading of Recommendations Assessment, Development, and Evaluation (GRADE) system was used to assess the risk of bias and quality of evidence from the included studies [11]. The level of evidence across all studies was ranked using this comprehensive assessment, providing an overall quality score that indicated a risk of bias. The Newcastle–Ottawa Scale was used to assess bias in retrospective and prospective cohort studies (Appendix B) [12]. We also evaluated the bias risk of RCTs using its updated Cochrane Risk of Bias tool for randomized trials (RoB 2) (Appendix D) [13]. The nonrandomized studies included in this review were also assessed using the MINORS tool (Appendix C) [14]. Such evaluations inform the reader about the quality of studies included and potential sources of bias, thereby increasing the trustworthiness of the results reported.

### 2.3. Data Synthesis

A meta-analysis could not be conducted because of major heterogeneity and the high inconsistency in data formats. This heterogeneity was further demonstrated by the diversity of study designs, such as RCTs, cohort studies, or observational studies, each using different forms of outcome assessments following surgery. The studies varied in terms of the demographics of the populations studied, such as the defined gestational age (term versus preterm neonates), birth weight, and comorbid conditions. These would impact surgical outcomes directly and would complicate inter-study comparisons. The surgical tactics for hernia repair (open vs. laparoscopic) and types of anesthesia (general vs. regional) were heterogeneous among the studies. Such differences influenced main features such as recurrence rates, complications, and recovery times, which have led to further inconsistencies. Outcomes, such as complication rates, recurrence rates, and postoperative outcomes, were inconsistently reported. Outcomes in some studies were assessed by the standardized scales, while others used descriptive metrics or did not quantify observations. These variations prevented us from synthesizing results into a single meta-analysis. Thus, we performed a basic descriptive statistical analysis with Review Manager version 5.4.1 (Cochrane, London, UK). This had the advantage of enabling us to report the results in an organized way while recognizing the concepts inherent in the data heterogeneity. Although heterogeneity across studies limited the applicability of some findings, it also highlighted the complexity in decision-making surrounding optimal surgical timing. Variability in the results underscores the importance of future studies employing standardized procedures and outcome measures. These modalities will also facilitate comparability across studies, resulting in a more solid evidential basis for clinical decision-making.

## 3. Results

A total of 7130 articles were identified during the initial database search. Then, we removed 5863 duplicates and applied the inclusion and exclusion criteria. Of these, 962 articles were excluded for reasons of availability of full-text, duplication, methodological rigor, studies that involved other surgical procedures or did not include a study analysis on the topic of neonatal inguinal hernias. Finally, articles not written in English and articles that contained inadequate data to determine the timing of surgery and the effect on prognosis were also excluded. This leaves 305 articles that were further assessed for eligibility. A total of 77 articles met all criteria for inclusion in this systematic review regarding the impact of surgical timing on outcomes in neonatal inguinal hernia repairs and underwent further evaluation [15,16,17,18,19,20,21,22,23,24,25,26,27,28,29,30,31,32,33,34,35,36,37,38,39,40,41,42,43,44,45,46,47,48,49,50,51,52,53,54,55,56,57,58,59,60,61,62,63,64,65,66,67,68,69,70,71,72,73,74,75,76,77].

The included papers contained a variety of study designs, including randomized controlled trials, cohort studies, and observational studies. This heterogeneity enabled a more in-depth assessment regarding the impact of the timing of surgical intervention on neonatal outcomes following the repair of an inguinal hernia. Figure 1 outlines the PRISMA flowchart demonstrating the selection process. All studies were published between 1970 and 2024, and extracted studies came from various parts of the world.

The articles referencing the effect of surgical timing on outcomes in neonatal inguinal hernia repairs, as well as characteristics of their cohort, are shown in Table 1.

### 3.1. Patients’ Profiles and Characteristics

The systematic review included a total of 96,674 patients undergoing inguinal hernia repairs. Out of these, 59,617 patients underwent early repairs, and 37,057 underwent delayed repairs. The ages of study participants varied considerably, ranging from neonates (<40 weeks gestational age to 28 days of life) to infants (29 days to 1 year of life), reflecting the diverse surgical timing practices and patient demographics.

The studies focusing on neonatal inguinal hernia repair not only point towards the devastating effect the condition holds but also surround the predominantly observed phenomenon, that is, males with significantly more representation than females. Based on a pooled analysis of studies, a male-to-female ratio of about 4:1 in neonatal and childhood populations was reported [44]. The results of our review were alongside a cohort study of 8037 infants assessing outcomes related to the timing of hernia repair, of which 3230 patients had early repair, thus demonstrating increased complications within the premature population, specifically recurrence and apnea [36]. Likewise, a comparative analysis of the timing of repair revealed that early repair significantly reduced the incarceration rates but also carried an increased risk of complications as well as increased length of stay, particularly in preterm infants [45]. These findings are consistent with the balance required between surgical urgency and the risks associated with early intervention in the preterm and neonatal populations. In addition, one study reported surgical techniques used, distinguishing between open and laparoscopic repairs. Patients undergoing laparoscopic repair had shorter hospitalizations and lower recurrence rates, especially in preterm neonates. However, it was also noted to require advanced expertise and facilities [28]. Another study compared the types of anesthesia, with general anesthesia being the most common approach [50]. Some studies have shown that the incidence of postoperative apnea was greater among preterm neonates, necessitating close monitoring during and after surgery [50,51]. There are limited studies reporting anesthesia’s long-lasting neurodevelopmental effects, underscoring the need for further research in this area [58]. Such numbers highlight the necessity to customize treatment and timing of intervention according to the patient to maximize the benefits in repair of the neonatal inguinal hernia.

### 3.2. Patient-Reported Outcomes and Complications

This section presents the key evidence related to patient outcomes and complications with neonatal and infant inguinal hernia repair, as shown in Table 2. An analysis was performed of six studies on preterm infants diagnosed with inguinal hernia during admission to the NICU. Early repair, performed pre-discharge was associated with significantly lower odds of hernia incarceration (OR 0.43, 95% CI 0.34–0.55, *p* < 0.001). However, it was also associated with increased postoperative pulmonary complications (OR 4.36, 95% CI 2.13–8.94, *p* < 0.001) and recurrence rates (OR 3.10, 95% CI 0.90–10.64, *p* = 0.07), while other complications of surgery (OR 0.94, 95% CI 0.18–4.83, *p* = 0.94) were similar in both the neonatal and infant groups. The study evidences a nuanced balance between minimizing the chances of incarceration by repairing early vs. limiting airway complications with delayed repair, underscoring that care should be taken to avoid both outcomes, and reiterates the importance of patient-tailored decisions considering factors such as age, weight, and comorbidities [46].

## 4. Discussion

We conducted a systematic review to evaluate the effect of the timing of surgical intervention on outcomes for neonates with inguinal hernias. Early and delayed surgical options both have unique risks and benefits that impact the final outcome of the procedure, as highlighted by our analysis. The timing of surgery is the most critical determinant of the probability of complications such as incarceration, recurrence, and postoperative respiratory morbidity, particularly in the cases of premature and critically ill neonates.

In our review, we demonstrated that early hernia repair, particularly if performed prior to discharge, greatly reduces the rates of incarceration and intestinal strangulation [55,67]. This is extremely significant in preterm newborns, who are at an increased risk of difficulties as a result of immature physiological development. However, early repair was also associated with postoperative complications, including respiratory distress and apnea, which are particularly worrisome for preterm infants [50,68]. As a result, these neonates may need prolonged hospitalization and further observation to address these problems appropriately [69]. Conversely, delaying surgery until after the infant has had time to grow and clinically stabilize minimizes the risk of early postoperative complications, including respiratory complications [53]. However, delaying the inguinal hernia repair comes with a higher chance of incarceration, which can cause complications such as bowel obstruction or strangulation [55,70]. Making such decisions requires balancing the benefit of allowing the infant some time to mature and stabilize against the risk of facing complications during or after surgery [71,72]. Importantly, comorbid conditions such as chronic lung disease and anemia were significantly associated with the outcomes, with patients with these other conditions more likely to suffer postoperative apnea and prolonged recovery [51,73]. It emphasizes the necessity for a patient-focused approach to timing choices, as the overall health and stability of the neonate should be determined along with the portion of early and postponed surgery [74].

Surgeons operating on neonates and infants with inguinal hernias are faced with a choice of surgical technique, and this is one of the more important decisions, as selecting a technique that accomplishes adequate repair without undue complications has a significant impact on outcomes. Laparoscopic repair demonstrated advantages such as lower rates of recurrence and decreased lengths of hospital stay. However, its success depends on the surgeon’s skill and institutional support. Alternatively, open repair is a proven technique that is available in all centers but is known to have longer recovery times in some cases [22]. Regarding anesthesia, general anesthesia was mostly used, but awake caudal anesthesia was found to reduce the risk associated with general anesthesia, especially in preterm neonates [73]. While studies have reported a safe profile of general anesthesia for most neonates, the neurodevelopmental effects of early exposure to anesthesia at such an early age remain to be determined [58]. Randomized controlled trials are needed to better assess these risks and to help establish age- and technique-based guidelines on its use.

Our results highlight the need for individualized treatment strategies guided by patient-specific factors such as gestational age, comorbidities, and current clinical status. Although early repair confers the most optimal reduction in incarceration risk, delayed repair may be the safer alternative in specific population groups, especially those who are preterm born or possess large coexisting comorbidities [49,75]. Additionally, this review indicates that there is no single ideal approach, and the decision to undergo early versus delayed surgical intervention should be made by the treating team of healthcare professionals, including the pediatric surgeon and neonatologist, considering the overall prognosis of the patient. More studies, in particular multicenter randomized controlled trials, are needed to define the optimal timing strategies of repair and provide stronger evidence to direct the clinical practice of neonatal inguinal hernia repair.

### Limitations

There are some limitations to this systematic review that need to be considered. Studies varied substantially concerning the surgical intervention used (open versus laparoscopic repair), anesthesia techniques, patient population characteristics, and methods of outcome assessment. Such heterogeneity could have affected the homogeneity of our results and precluded a meta-analysis, thus preventing a robust conclusion. Moreover, the majority of articles did not provide full perioperative care protocols, and, thus, specific operation-related aspects (recovery times, complications, etc.) could not be elucidated. Another major limitation is the inability to control for confounders that may influence surgical outcomes. Disparities in access to care, including subspecialty referral time, subspecialty pediatric surgeon availability, and hospital resources, may have affected the outcomes reported. Biased variability of recurrence rates and complications could also have been introduced due to the varying degrees of experience of the surgical team and the option for advanced laparoscopic techniques. These factors were not uniformly reported or adjusted for between studies, making it difficult to analyze the impact of surgical timing in isolation.

Moreover, the long-term effect of anesthesia on the neurodevelopment of neonates remains poorly characterized, which is an important consideration when weighing the risks of early versus delayed repair. The variation in follow-up reporting on these outcomes highlights the need for stronger, standardized reporting. To fill these gaps, further prospective, large-scale, multicenter trials with standardized parameters are needed in future studies. Standardized protocols regarding patient selection, surgical techniques, and postoperative care will improve comparability and yield better evidence for clinical practice guidelines. Additionally, studies should include socioeconomic and healthcare access-related factors to address uneven outcomes and promote equity in care across populations. Lastly, well-designed long-term follow-up studies are required to assess neurodevelopmental and quality-of-life outcomes between the various approaches to surgery timing.

## 5. Conclusions

This systematic review highlights that the timing of inguinal hernia repair in neonate and infant periods significantly influences the outcomes. Early repair (the first 28 days of life) reduces the risk of incarceration but increases the risks of postoperative complications, especially in preterm neonates. An individualized approach based on patient-specific factors such as age, comorbidities, and surgical resources is necessary for optimizing outcomes. We need additional randomized controlled trials to establish evidence-based guidelines for surgical timing.

## Figures and Tables

**Figure 1 pediatrrep-17-00012-f001:**
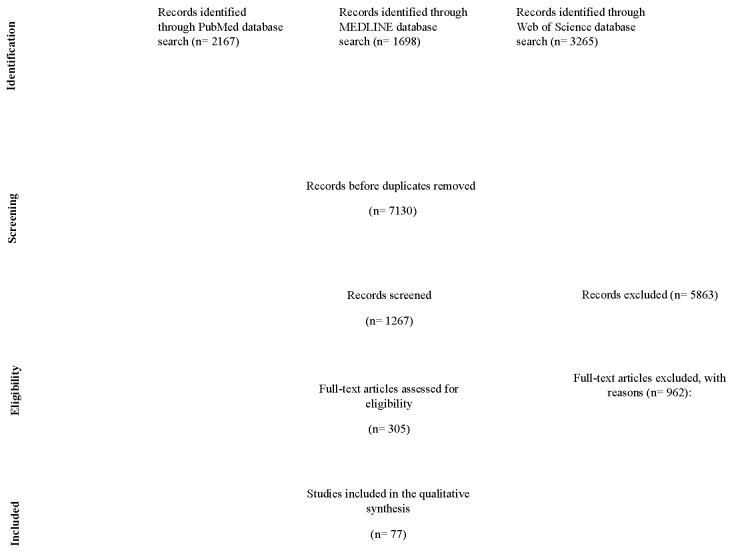
Detailed PRISMA chart used for this systematic review, outlining the many stages of this study’s selection process.

**Table 1 pediatrrep-17-00012-t001:** Characteristics and outcomes of studies investigating impact of surgical timing on outcomes in neonatal inguinal hernia repairs.

Authors	Country	Study Design	Patients (N)	Age	Summary	Level of Evidence
Masoudian P [15]	Canada	Systematic review and meta-analysis	1761	<1 year	Repairing an IH neonatal (3–6 months) in premature infants decreases the risk of complications compared to infant repair (>6 months), though this may come at the cost of increased recurrence.	III
Khasawneh W [16]	Jordan	Retrospective cohort study	272	Median: 49 weeks	Repair is safe if performed electively within seven days of diagnosis and does not lead to incarceration or postoperative apnea.	III
Choo CSc [17]	Singapore	Retrospective cohort study	219	Median: Neonatal group (37.7 weeks); infant group (42.1 weeks)	Delaying herniotomy post-discharge is safe under careful monitoring, and it may resolve spontaneously.	III
Kus N [18]	Philadelphia	Retrospective cohort study	836	<1 year	The time to repair is significantly affected by the type of insurance and systemic issues, and a longer duration is associated with more healthcare visits.	III
Soyer T [19]	Europe	Cross-sectional study	180	<32 weeks gestation	Practices for the timing of IH repair vary considerably, although most surgeons prefer pre-discharge repair to minimize incarceration risk, but some delay to minimize the risk of apnea.	IV
Cho YJ [20]	South Korea	Multicenter retrospective study	149	<37 weeks gestation	Repair of congenital IH delayed until NICU discharge is associated with fewer recurrences and postoperative respiratory insufficiency compared to neonatal repair.	III
Crankson S [21]	Saudi Arabia	Retrospective chart review	84	Group 1: Median 39.5 weeks; Group 2: median 66.5 weeks	Repair after discharge from NICU allows for the repair closer to term without the increase in incarceration risk.	III
Walsh CM [22]	USA	Retrospective cohort study	80	<12 months	Regarding the timing of LIHR, there was no difference in recurrence rates, but further preoperative and postoperative management may be needed in infants ≤3 months, especially concerning anemia and comorbidities.	III
Prato AP [23]	Italy	Retrospective cohort study	154	Median: 42 months	Postoperative follow-up of very low birth weight (weight < 5 kg) infants is essential due to an increased risk of metachronous hernia.	III
Castro BA [24]	Spain	Retrospective cohort study	156	≤6 months	LIHR in preterm infants is safe and effective, with outcomes similar to those of preterm infants.	III
Peace AE [25]	USA	Retrospective cohort study	3662	<37 weeks GA	Neonatal IHR in the preterm infant saves costs but at the expense of a higher risk of recurrence.	III
Haveliwala Z [26]	USA	Retrospective cohort study	1195	GA: 31 weeks to 14.6 years	LIHR is both effective and safe across all ages. Outcomes are improved using a double stitch technique, particularly in preterm babies.	III
Verhelst J [27]	The Netherlands	Retrospective cohort study	132	<3 months after birth	Elective IHR in premature infants approaches a lower cost compared to emergency IHR and should be prioritized if clinically appropriate to reduce healthcare costs.	III
Safa N [28]	Canada	Retrospective comparative study	1952	<1 year	LIHR offers a modest benefit in decreasing metachronous hernias associated with markedly increased recurrence rates over open IHR.	III
Bawazir OA [29]	Saudi Arabia	Prospective cohort study	118	Neonates and preterm infants (weight < 5 kg)	Delaying elective hernia repair for neonates and preterm infants is associated with reduced risk of preoperative and postoperative complications in this patient population.	II
Ramsey WA [30]	USA	Retrospective cohort study	6148	<1 year	Selective deferred repair of incarcerated inguinal hernia after successful manual reduction appears to be safe and reduces complications.	III
Akinkuotu AC [31]	USA	Retrospective cohort study	30298	<37 weeks gestation, ≤28 days old at admission	Neonatal IHR is affected by patient age group, hospital type, and additional procedures.	III
Olesen CS [32]	Denmark	Cross-sectional study	48	<1 year	Patients with pediatric inguinal hernias have inconsistency in the timing of surgical management; therefore, there is a need for standardized guidelines to improve education and ultimately outcomes for those patients.	IV
Fukuhara M [33]	Japan	Retrospective cohort study	61	<3 months	Neonatal IH could be SR, and elective surgery can be safely delayed until after 6 months to avoid risks and prevent unnecessary neonatal surgeries.	III
Youn JK [34]	Australia	Retrospective cohort study	205	Very low birth weight infants (<1500 g)	Infants with VLBW, specifically male children, are at high risk for inguinal hernias, not only during infancy but until 8 years of life.	III
Taylor K [35]	USA	Retrospective cohort study	9993	<1 year	Children with more than one comorbidity had a high risk of hernia recurrence, indicating that they require close follow-up.	II
Gulack BC [36]	USA	Multicenter cohort study	8037	<34 weeks gestation	IHR for premature infants is affected by socioeconomic and clinical characteristics.	III
Wolf LL [37]	USA	Retrospective cohort study	19398	<1 year	The study reveals that there is a disparity in the timing of repairs depending on hernia type and highlights the need for evidence-based guidelines to optimize surgical timing.	II
Demouron M [38]	France	Retrospective cohort study	55	<6 months	There is no indication of routine contralateral exploration during unilateral IHR.	III
Sonderman KA [39]	USA	Retrospective cohort study	8897	Median: 2 years	Inguinal hernia repair is complicated by testicular atrophy as a rare complication. The greatest risk is in boys <2 years and males with undescended testis.	II
Dreuning KM [40]	Netherlands	Retrospective cohort study	1084	Median: 133.5 weeks	Delaying IHR increases the risk of hernia strangulation, especially in young gestation and high birth weight.	III
Shalaby R [41]	Egypt	Retrospective cohort study	1284	6 to 78 months	LIHR was found to be safe and effective for all ages; it also allows the repair and detection of contralateral hernia and all types of IHs.	III
Kwasau H [42]	Sierra Leone	Retrospective cohort study	215	<1 year	Untreated IH increases the risk of complications and mortality.	IV
Alhalabi R [43]	United Arab Emirates	Case report study	1	3 months	Neonatal IHR is important to prevent developing colonic stricture and mortality.	V

Abbreviations: IH: inguinal hernia; NICU: neonatal intensive care unit; LIHR: laparoscopic inguinal hernia repair; GA: gestational age; IHR: inguinal hernia repair; SR: spontaneous reduction; VLBW: very low birth weight.

**Table 2 pediatrrep-17-00012-t002:** Comparison of complications in neonatal vs. infant inguinal hernia repair.

Authors	Outcomes	Early Repair (During Neonatal Period: 0–28 Days of Life)	Delayed Repair (During Infant Period: 29 Days to 1 Year of Life)	*p* Value	Interpretation
Sharma P [47]	Cardiac Arrest	0%	3%	<0.05	Rare but seen in the infant group.
Ferrantella A [48]	Readmission within 1 year	35%	18%	<0.001	The rate of unplanned readmissions was higher in preterm and low full-term infants.
Rescorla FJ [49]	Incarceration	31%	8%	Not provided	The percentage of incarceration was higher in preterm and critically ill infants.
Cote CJ [50]	Postoperative apnea	High risk	Low risk	<0.0001	Early repair is associated with a higher apnea rate in preterm infants, which requires surveillance and management of anesthesia accordingly.
Welborn LG [51]	Postoperative apnea	79%	21%	<0.03	Preterm infants with anemia have a high risk of developing postoperative apnea.
Stylianos S [52]	Bowel obstruction	9.4%	3%	Not provided	It is important to schedule elective hernia repair surgery as soon as possible to reduce incarceration risk and complications associated with incarceration.
Kumar VHS [53]	Chronic lung disease	High risk	Low risk	<0.05	Preterm infants have a higher risk of developing chronic lung disease compared to full-term babies.
Mowitz ME [54]	Readmission within 1 year	High risk	Low risk	<0.05	Early repair in extremely premature neonates is associated with a higher risk of readmission within one year compared to delayed repair.
Lautz TB [55]	Incarceration	11%	21%	0.002	Delaying hernia repair after discharge (>40 weeks) increases the risk of incarceration.
Sacks MA [56]	Blood transfusions	4.2%	0.5%	0.036	Blood transfusions are more common in preterm infants, indicating their fragility.
Sulkowski JP [57]	Recurrence	5.9%	3.7%	0.02	The risk of recurrence is slightly higher with neonatal repair, potentially as a result of immature healing of the tissue.
Blakely ML [58]	Neurodevelopmental impairment	69%	70%	0.03	There was no overall difference in neurodevelopmental impairment rates between the two groups.
Canadian Association [59]	Incarceration	High risk	Low risk	Not provided	Infants less than 1-year-old should have inguinal hernia repair early to avoid incarceration.
Rajput A [60]	Incarceration	High risk	Low risk	Not provided	Preterm infants are more prone to complications such as irreducibility or incarceration, which necessitate relief and repair as early as possible.
Harper RC [61]	Cardiac arrest	High risk	Low risk	Not provided	Extremely low birth weight infants (≤1000 g) have a higher risk of developing postoperative complications, so they need careful preoperative planning.
Rowe MI [62]	Mortality	Low risk	High risk	Not provided	Delaying inguinal hernia repair increases the risk of complications and mortality.
Puri P [63]	Testicular atrophy	2.3%	6.8%	Not provided	Testicular atrophy is a rare complication but can occur; early intervention can minimize this risk.
Uemura S [64]	Incarceration	4.8%	10.5%	Not provided	Delaying inguinal hernia repair is associated with a risk of complications like incarceration and strangulation.
Baird R [65]	Recurrence	9.7%	5.2%	<0.005	Preterm infants have a high risk of developing vas injury, recurrence, testicular atrophy, wound infection, and hydrocele.
Borenstein SH [66]	Mortality	1%	1.10%	Not provided	There is no big difference in mortality rate between neonatal and infant inguinal hernia repair.

## Data Availability

Not applicable.

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
