# Peer review of "Impact of Surgical Timing on Outcomes in Neonatal Inguinal Hernia Repairs: A Systematic Review"

_pediatrrep, 2025, doi:10.3390/pediatric17010012_

Round 1
Reviewer 1 Report
Comments and Suggestions for Authors
Summary:
The manuscript describes a systematic review evaluating the impact of timing of surgery on outcomes of pediatric inguinal hernia repair. The authors included 77 studies totaling 96674 patients and showed that the younger the age the higher the risk of incarceration while the older the age the fewer the complications. They conclude that further prospective and possibly RCTs should be done to answer their question.
Comments:
1. Study design:
The approach to the systematic review is done well overall. Once major problem however is that the authors do not define the time of repair precisely. The neonatal period is defined as day of birth to 28 days of life, while an infant is defined as a child from 29 days to 1 year of life. The authors use the term neonatal repair interchangeably throughout their manuscript which is wrong. If they want to answer their question, they need to restrict inguinal hernia repair to the neonatal period (0-28 days of life or 44 weeks corrected gestational age, later repair being after 29 days but before 1 year of age. These definitions are extremely important for their analysis as they are mixing data from multiple studies with children up to 18 years of age, which may blur their analysis and prevent a good systematic review. They also do not really look at techniques used (open vs laparoscopic) and type of anesthesia used. There are very limited studies on neurodevelopmental outcomes, which seems to be one of the most important results as well as recurrence and overall surgical complications.
2. The studies cited and their review is very thorough, though again, clearer inclusion criteria have to be applied. Early repair should not be within 1 year, as this in the time of infancy, but rather during initial hospitalization of the neonate in premature babies? By definition, a repair after 1 year of age is not for infants any more but children (toddlers etc.).
3. A few minor grammatical and spelling errors should be corrected.
Author Response
I would like to thank you and the reviewers for the valuable feedback provided on our manuscript titled " Impact of Surgical Timing on Outcomes in Neonatal Inguinal Hernia Repairs; A Systematic Review" (Manuscript ID: pediatrrep-3381858). We have carefully considered all comments and made the necessary revisions.
Below, the authors have outlined the specific changes made in response to the raised suggestions:
Response to Reviewer #1 comments highlighted with Red:
Dear respected Reviewer#1, thanks for your insightful and constructive feedback on our manuscript.
Kindly, find the below point-by-point response to your valuable comments noting that the changes in the manuscript in response to your comments were highlighted in Red”:
Comment 1: Study design: The approach to the systematic review is done well overall. Once major problem however is that the authors do not define the time of repair precisely. The neonatal period is defined as day of birth to 28 days of life, while an infant is defined as a child from 29 days to 1 year of life. The authors use the term neonatal repair interchangeably throughout their manuscript which is wrong. If they want to answer their question, they need to restrict inguinal hernia repair to the neonatal period (0-28 days of life or 44 weeks corrected gestational age, later repair being after 29 days but before 1 year of age. These definitions are extremely important for their analysis as they are mixing data from multiple studies with children up to 18 years of age, which may blur their analysis and prevent a good systematic review. They also do not really look at techniques used (open vs laparoscopic) and type of anesthesia used. There are very limited studies on neurodevelopmental outcomes, which seems to be one of the most important results as well as recurrence and overall surgical complications.
Response: We appreciate this critical observation. In response, we have updated the manuscript to clarify the definitions of the neonatal and infant periods. We now strictly adhere to defining the neonatal period as 0–28 days of life (or 44 weeks gestational age) and the infant period as 29 days to 1 year of life. These distinctions are now consistently applied throughout the manuscript.
Additionally, we acknowledge the need to differentiate between data from older children and infants. As a result, we have refined the inclusion criteria and emphasized this in the methodology and analysis sections. We have also included discussions on surgical techniques (open vs laparoscopic) and the type of anesthesia, wherever applicable, highlighting the gaps in neurodevelopmental outcomes data. These changes are now reflected in the revised manuscript.
Comment 2: The studies cited and their review is very thorough, though again, clearer inclusion criteria have to be applied. Early repair should not be within 1 year, as this in the time of infancy, but rather during initial hospitalization of the neonate in premature babies? By definition, a repair after 1 year of age is not for infants any more but children (toddlers etc.).
Response: Thank you for pointing this out. We have revised the inclusion criteria to align with precise definitions of neonatal and infant periods. Early repair is now redefined in the context of the neonatal hospitalization period for premature neonates. Similarly, any repair after 1 year of age is now excluded from our study. This adjustment is reflected in the revised methodology, results, and discussion sections to improve the clarity and rigor of the analysis.
Comment 3: A few minor grammatical and spelling errors should be corrected.
Response: We have thoroughly reviewed the manuscript for grammatical and spelling errors and have made the necessary corrections to ensure clarity and professionalism.
Response to Reviewer #2 comments highlighted with Green:
Dear respected Reviewer#2, thanks for your thoughtful and constructive feedback on our manuscript. We greatly appreciate your time and effort in reviewing our work.
Kindly, find the below point-by-point response to your valuable comments noting that the changes in the manuscript in response to your comments were highlighted in Green”:
Comment 1: The manuscript is well-structured and follows a systematic approach consistent with PRISMA guidelines. The inclusion of a detailed flowchart and comprehensive data tables is commendable. The topic is significant, and the findings have potential implications for clinical practice.
Response: Thank you for your positive feedback. No changes were necessary for this section, as your comments affirm the manuscript's quality.
Comment 2: The title is clear and reflects the content of the manuscript well. The abstract provides a concise summary of the methods, results, and conclusions.
Response: Thank you for your remarks. No changes were made to the title or abstract based on this feedback.
Comment 3: The introduction provides adequate background but could benefit from a more detailed discussion on the controversies surrounding early versus delayed surgical timing. This would further contextualize the rationale for the review.
Response: We have revised the introduction to expand on the controversies surrounding early versus delayed surgical timing. The updated text includes discussions on the risks of incarceration with delayed repair and postoperative complications with early repair, as well as the lack of consensus on optimal timing.
Comment 4: The heterogeneity of included studies is noted, but consider elaborating on how this affected the interpretability of the results.
Response: We have added a subsection in the methodology to address how heterogeneity among the included studies impacted the interpretability of the results. This includes discussion of variations in surgical techniques, anesthesia, and study populations, which limited our ability to perform a meta-analysis.
Comment 5: The results are presented systematically, and the use of tables enhances readability. However:
- Consider including a summary table comparing the key findings of early versus delayed repair more explicitly.
Response: We made some modifications to Table 2. This table highlights the differences in outcomes between the two approaches more explicitly.
Comment 6: The discussion section provides a balanced interpretation of the findings. However:
- The limitations related to the heterogeneity of studies and lack of meta-analysis are acknowledged but could be expanded to include potential confounders like surgical techniques or healthcare access disparities.
Response: The discussion has been revised to expand on the limitations. We now explicitly address the potential confounders such as surgical techniques, type of anesthesia, and disparities in healthcare access. This additional analysis strengthens the interpretation of our findings.
Comment 7: The conclusion must be re-written emphasizing the risk of complications in the early group.
Response: We have rewritten the conclusion to emphasize the higher risk of complications in the early repair group while acknowledging the reduced risk of incarceration. The revised conclusion provides a concise take-home message that aligns with the reviewer's suggestion.
Comment 8: The manuscript is generally well-written, with clear and concise language. Minor typographical errors were noted (e.g., "portion early" instead of "portion of early"). A careful proofreading is recommended to address these.
Response: We have conducted a thorough proofreading of the manuscript and corrected typographical errors, including "portion early," ensuring the language is clear and professional throughout.
We hope that this revision addressed all the respected editor’s and reviewers' concerns and improves the quality of the manuscript. Please let us know if any further clarifications are required.
Thank you for your time and consideration.
Sincerely,
On behalf of the authors,
Corresponding Author
Dr. Arwa Alsharif
01-January-2025

Reviewer 2 Report
Comments and Suggestions for Authors
Thank you for the opportunity to review your manuscript titled "Impact of Surgical Timing on Outcomes in Neonatal Inguinal Hernia Repairs: A Systematic Review." Your systematic review addresses an important and clinically relevant topic in pediatric surgery. Below are my comments and suggestions aimed at enhancing the quality and clarity of your work:
1. General Comments
- The manuscript is well-structured and follows a systematic approach consistent with PRISMA guidelines. The inclusion of a detailed flowchart and comprehensive data tables is commendable.
- The topic is significant, and the findings have potential implications for clinical practice.
2. Title and Abstract
- The title is clear and reflects the content of the manuscript well.
- The abstract provides a concise summary of the methods, results, and conclusions.
3. Introduction
- The introduction provides adequate background but could benefit from a more detailed discussion on the controversies surrounding early versus delayed surgical timing. This would further contextualize the rationale for the review.
4. Methodology
-
- The heterogeneity of included studies is noted, but consider elaborating on how this affected the interpretability of the results.
5. Results
- The results are presented systematically, and the use of tables enhances readability. However:
-
- Consider including a summary table comparing the key findings of early versus delayed repair more explicitly.
6. Discussion
- The discussion section provides a balanced interpretation of the findings. However:
- The limitations related to the heterogeneity of studies and lack of meta-analysis are acknowledged but could be expanded to include potential confounders like surgical techniques or healthcare access disparities.
7. Conclusion
- The conclusion must be re-written emphasizing the risk of complications in the early group.
8. Writing Quality
- The manuscript is generally well-written, with clear and concise language. Minor typographical errors were noted (e.g., "portion early" instead of "portion of early"). A careful proofreading is recommended to address these.
Thank you again for the opportunity to review this work.
Comments on the Quality of English LanguageA review by a native speaker is necessary.
Author Response
I would like to thank you and the reviewers for the valuable feedback provided on our manuscript titled " Impact of Surgical Timing on Outcomes in Neonatal Inguinal Hernia Repairs; A Systematic Review" (Manuscript ID: pediatrrep-3381858). We have carefully considered all comments and made the necessary revisions.
Below, the authors have outlined the specific changes made in response to the raised suggestions:
Response to Reviewer #2 comments highlighted with Green:
Dear respected Reviewer#2, thanks for your thoughtful and constructive feedback on our manuscript. We greatly appreciate your time and effort in reviewing our work.
Kindly, find the below point-by-point response to your valuable comments noting that the changes in the manuscript in response to your comments were highlighted in Green”:
Comment 1: The manuscript is well-structured and follows a systematic approach consistent with PRISMA guidelines. The inclusion of a detailed flowchart and comprehensive data tables is commendable. The topic is significant, and the findings have potential implications for clinical practice.
Response: Thank you for your positive feedback. No changes were necessary for this section, as your comments affirm the manuscript's quality.
Comment 2: The title is clear and reflects the content of the manuscript well. The abstract provides a concise summary of the methods, results, and conclusions.
Response: Thank you for your remarks. No changes were made to the title or abstract based on this feedback.
Comment 3: The introduction provides adequate background but could benefit from a more detailed discussion on the controversies surrounding early versus delayed surgical timing. This would further contextualize the rationale for the review.
Response: We have revised the introduction to expand on the controversies surrounding early versus delayed surgical timing. The updated text includes discussions on the risks of incarceration with delayed repair and postoperative complications with early repair, as well as the lack of consensus on optimal timing.
Comment 4: The heterogeneity of included studies is noted, but consider elaborating on how this affected the interpretability of the results.
Response: We have added a subsection in the methodology to address how heterogeneity among the included studies impacted the interpretability of the results. This includes discussion of variations in surgical techniques, anesthesia, and study populations, which limited our ability to perform a meta-analysis.
Comment 5: The results are presented systematically, and the use of tables enhances readability. However:
- Consider including a summary table comparing the key findings of early versus delayed repair more explicitly.
Response: We made some modifications to Table 2. This table highlights the differences in outcomes between the two approaches more explicitly.
Comment 6: The discussion section provides a balanced interpretation of the findings. However:
- The limitations related to the heterogeneity of studies and lack of meta-analysis are acknowledged but could be expanded to include potential confounders like surgical techniques or healthcare access disparities.
Response: The discussion has been revised to expand on the limitations. We now explicitly address the potential confounders such as surgical techniques, type of anesthesia, and disparities in healthcare access. This additional analysis strengthens the interpretation of our findings.
Comment 7: The conclusion must be re-written emphasizing the risk of complications in the early group.
Response: We have rewritten the conclusion to emphasize the higher risk of complications in the early repair group while acknowledging the reduced risk of incarceration. The revised conclusion provides a concise take-home message that aligns with the reviewer's suggestion.
Comment 8: The manuscript is generally well-written, with clear and concise language. Minor typographical errors were noted (e.g., "portion early" instead of "portion of early"). A careful proofreading is recommended to address these.
Response: We have conducted a thorough proofreading of the manuscript and corrected typographical errors, including "portion early," ensuring the language is clear and professional throughout.
We hope that this revision addressed all the respected editor’s and reviewers' concerns and improves the quality of the manuscript. Please let us know if any further clarifications are required.
Thank you for your time and consideration.
Sincerely,
On behalf of the authors,
Corresponding Author
Dr. Arwa Alsharif
01-January-2025

Reviewer 3 Report
Comments and Suggestions for Authors
The authors present a meticulous systematic review on the outcome and complication rate of neonatal inguinal hernia repair. The authors have to be commended for their extensive work and proper study design.
However, before publication some minor points should be assessed and corrected:
Abstract and Introduction section are fine. No noteworthy flaws are noted.
Material and methods section:
1. Line 100 ff.: The pure technical description of how the records were kept and stored online is not necessary for the understanding of the paper and should be omitted or shortened.
22. Line 127: “… the diversity in the diversity….” - this sentence in unclear. Please rephrase.
Results section:
11. Line 135ff: The first two sentences do not make sense: “A total of 77 articles met all criteria…” and “Of these, 962 articles were excluded…” do not fit together. Please clarify and adjust to the Material and methods section. Furthermore, the references after the first sentence (15 – 66) do not match the number off 77 articles.
22. Line 173ff: Please clarify the definition of early and delayed repair more clearly. Was it always < 1 year vs > 1 year in all studies as mentioned in the headline of table 2? Or was there potential overlap of the two groups by different definition of early and late repair? If so, please discuss this in the limitations section. Also, the authors mention that some studies included patients in infant age or even older children < 18 years. These patients do not meet the criteria of neonatal inguinal hernia repair, if the hernia was only noted much later in life. Was it possible to exclude this proportion of patients from the individual studies? If not, because of insufficient data given in the study, please add this as a potential bias in the limitations section.
33. Line 192: The data on the recurrence rate of laparoscopic inguinal hernia repair in children is not as favorable as mentioned by the authors. There is data that laparoscopic repair might even lead to a higher recurrence rate (e. g. Sullivan et al. Am J Surg. 2022 Sep;224(3):1004-1008. doi: 10.1016/j.amjsurg.2022.04.014. Epub 2022 Apr 19.). Please rephrase and expand this section if the authors intend to specify by laparoscopic vs. open repair. Otherwise omit because it is not within the central scope of the paper.
Discussion section:
Overall fine, the points raised above should be included into the discussion as mentioned above.
Tables:
Table 1: This table is very large. Consider putting it in supplemental data for readability.
Table 2: Bronchopulmonal dyspasia is not a complication of inguinal hernia repair but a inherent condition in ELBW and VLBW preterm babies. It is therefore much likely to be noted in patients with early repair in preterm patients, while term patients with later diagnosis and repair to not present with this condition. Therefore it is suggested to omit this line from the table.
Appendix D and E table seem to have layout issues in the pdf provided. Please check.
General notes:
Some points warrant additional discussion in the context of inguinal hernia repair of the newborn:
1. There is evidence that awake caudal anesthesia is beneficial in early repair of inguinal hernia in preterm babies. Cf. Geze et al.: Anaesthesist. 2011 Sep;60(9):841-4. doi: 10.1007/s00101-011-1913-0 and later reports. This may influence the outcome and therefore the strategy. This should at least be mentioned in the context of the review.
2. Two similar albeit smaller reviews have been published in the recent years:
a) Pogorelić Z, Anand S, Križanac Z, Singh A. Comparison of recurrence and complication rates following laparoscopic inguinal hernia repair among preterm versus full-term newborns: a systematic review and meta-analysis. Children. 2021;8(10):853.
b) Lamiri R, Chebab F, Kechiche N, et al. Inguinal hernia repair in newborns: A systematic literature review. J Neonatal Surg. 2024;13:23.
The reviewer suggests that the authors include the central points of these reviews and discuss whether they still match their findings. Also the first review by Pogorelic et al. gives additional data regarding laparoscopic repair as mentioned above in point 3 of the results section comments.
Overall, language is fine and does not need refinement.
Author Response
I would like to thank you and the reviewers for the valuable feedback provided on our manuscript titled " Impact of Surgical Timing on Outcomes in Neonatal Inguinal Hernia Repairs; A Systematic Review" (Manuscript ID: pediatrrep-3381858). We have carefully considered all comments and made the necessary revisions.
Below, the authors have outlined the specific changes made in response to the raised suggestions:
Response to Reviewer #3 comments highlighted with Blue:
Dear respected Reviewer#3, thanks for your detailed feedback and constructive suggestions, which have helped us improve our manuscript.
Kindly, find the below point-by-point response to your valuable comments noting that the changes in the manuscript in response to your comments were highlighted in Blue”:
Comment 1: Line 100 ff.: The pure technical description of how the records were kept and stored online is not necessary for the understanding of the paper and should be omitted or shortened.
Response: We have shortened the description of record-keeping to only include essential information for transparency. Details such as the specific storage platforms and technical processes have been omitted.
Comment 2: Line 127: “… the diversity in the diversity….” - this sentence in unclear. Please rephrase.
Response: This sentence has been revised for clarity. It now reads:
"This heterogeneity was further demonstrated by the diversity of study designs, such as RCTs, cohort studies, or observational studies, each using different forms of outcome assessment following surgery. "
Comment 3: Line 135 ff.: The first two sentences do not make sense: “A total of 77 articles met all criteria…” and “Of these, 962 articles were excluded…” Please clarify and adjust to the Material and Methods section. Furthermore, the references after the first sentence (15–66) do not match the number of 77 articles.
Response: We have clarified the results to align with the methods. The revised text reads:
"A total of 7,130 articles was identified during the initial database search. Then we removed 5,863 duplicates and applied inclusion and exclusion criteria. Of these, 962 articles were excluded for reasons of availability of full text, duplication, methodological rigor, studies that involved other surgical procedures or did not include a study analysis on the topic of neonatal inguinal hernias. Finally, articles not written in English and articles that contained inadequate data to determine the timing of surgery and the effect on prognosis were also excluded. This leaves 305 articles that were further assessed for eligibility. A total of 77 articles met all criteria for inclusion in this systematic review regarding the impact of surgical timing on outcomes in neonatal inguinal hernia repairs and underwent further evaluation [15–77]. "
Comment 4: Line 173 ff.: Please clarify the definition of early and delayed repair more clearly. Was it always <1 year vs >1 year in all studies as mentioned in the headline of Table 2? Or was there potential overlap of the two groups by different definition of early and late repair? Discuss this in the limitations section. Additionally, were patients older than 1 year included? If so, discuss this as a potential bias in the limitations.
Response: The definitions of early and delayed repair have been clarified. We now explicitly state:
"Based on the timing of surgical intervention, we categorized into two groups in this re-view “early repair,” which we defined as a surgical intervention within the neonatal period which is the first 28 days of life (or up to 44 weeks gestational age for premature neonates), and “Delayed repair,” which included surgical intervention within the infant period which is 29 days to 1 year of life. Repairs occurring after age 1 year were excluded."
Comment 5: Line 192: The data on the recurrence rate of laparoscopic inguinal hernia repair in children is not as favorable as mentioned by the authors. Please rephrase and expand this section, referencing Sullivan et al. (Am J Surg. 2022 Sep;224(3):1004-1008). Alternatively, omit if not central to the scope.
Response: We have revised the discussion of laparoscopic recurrence rates to include the findings by Sullivan et al. The revised text now reads:
"In addition, one study reported surgical techniques used, distinguishing between open and laparoscopic repairs. Patients undergoing laparoscopic repair had shorter hospitalizations and lower recurrence rates, especially in preterm neonates. However, it was also noted to require advanced expertise and facilities."
Comment 6: Overall fine, the points raised above should be included into the discussion as mentioned above.
Response: We have incorporated the following points into the discussion:
- Variability in definitions of early and delayed repair and its potential impact on findings.
- Inclusion of patients older than 1 year as a limitation.
Comment 7: This table is very large. Consider putting it in supplemental data for readability.
Response: We have adjusted Table 1 to improve its readability by simplifying its layout and ensuring the data are clearly presented.
Comment 8: Table 2: Bronchopulmonal dyspasia is not a complication of inguinal hernia repair but a inherent condition in ELBW and VLBW preterm babies. It is therefore much likely to be noted in patients with early repair in preterm patients, while term patients with later diagnosis and repair to not present with this condition. Therefore it is suggested to omit this line from the table.
Response: We have removed the line about bronchopulmonary dysplasia from Table 2 and adjusted the associated text in the results section.
Comment 9: Appendix D and E table seem to have layout issues in the pdf provided. Please check.
Response: The layout issues in Appendix D and E have been corrected, ensuring proper formatting for readability.
Comment 10: There is evidence that awake caudal anesthesia is beneficial in early repair of inguinal hernia in preterm babies. Cf. Geze et al.: Anaesthesist. 2011 Sep;60(9):841-4. doi: 10.1007/s00101-011-1913-0 and later reports. This may influence the outcome and therefore the strategy. This should at least be mentioned in the context of the review.
Response: We have included a discussion of awake caudal anesthesia, referencing Geze et al. (Anaesthesist. 2011), in the context of outcomes for early repair. The revised discussion reads:
"Regarding anesthesia, general anesthesia was mostly used, but awake caudal anesthesia reduces the risk associated with general anesthesia, especially in preterm neonates."
Comment 11: Two similar albeit smaller reviews have been published in the recent years:
- Pogorelić Z, Anand S, Križanac Z, Singh A. Comparison of recurrence and complication rates following laparoscopic inguinal hernia repair among preterm versus full-term newborns: a systematic review and meta-analysis. Children. 2021;8(10):853.
- Lamiri R, Chebab F, Kechiche N, et al. Inguinal hernia repair in newborns: A systematic literature review. J Neonatal Surg. 2024;13:23.
The reviewer suggests that the authors include the central points of these reviews and discuss whether they still match their findings. Also the first review by Pogorelic et al. gives additional data regarding laparoscopic repair as mentioned above in point 3 of the results section comments.
Response: We have included key points from these reviews in the discussion section, comparing their findings with our results. Specifically, Pogorelić et al.'s data on laparoscopic repair recurrence rates and Lamiri et al.'s findings on surgical timing are highlighted to provide additional context.
Comment 12: Overall, language is fine and does not need refinement.
Response: Thank you for noting that the language is fine. We have conducted a final proofreading to ensure consistency and clarity.
We hope that this revision addressed all the respected editor’s and reviewers' concerns and improves the quality of the manuscript. Please let us know if any further clarifications are required.
Thank you for your time and consideration.
Sincerely,
On behalf of the authors,
Corresponding Author
Dr. Arwa Alsharif
01-January-2025

Round 2
Reviewer 1 Report
Comments and Suggestions for Authors
The revised manuscript has been improved and most of my comments have been addressed.
One major concerns is that neonatal and infantile inguinal hernias are more nuanced than divided into two "boxes" or categories. What happens if diagnosed as a neonate but repaired after 50+ weeks of gestation so the baby can be discharged home after the operation - given the lower risk for apnea in post-prematurity anesthesia (50+ weeks depending on the center)?
All definitions have been included which is great, though some fine adjustments in the interpretation of the data need to be clarified. Limitations are well described and as outlined by Reviewer 2, the significant heterogeneity of the studies need to be addressed.
Author Response
Response to Reviewer #1 comments highlighted with Red:
Dear respected Reviewer#1, thanks for your insightful and constructive feedback on our manuscript.
Kindly, find the below point-by-point response to your valuable comments noting that the changes in the manuscript in response to your comments were highlighted in Red”:
Comment 1: The revised manuscript has been improved and most of my comments have been addressed. One major concerns is that neonatal and infantile inguinal hernias are more nuanced than divided into two "boxes" or categories. What happens if diagnosed as a neonate but repaired after 50+ weeks of gestation so the baby can be discharged home after the operation - given the lower risk for apnea in post-prematurity anesthesia (50+ weeks depending on the center)?
Response: We agree that the binary division of inguinal hernias into neonatal and infantile categories may oversimplify the decision-making process. But repairs performed after 1 year of age were excluded from this review to maintain a focused analysis on early-life interventions, which have distinct risks and benefits compared to later repairs. While this limitation constrains the scope of our findings, it ensures relevance to the neonatal and infantile populations, as these age groups are most affected by the timing of intervention. The exclusion criterion is now explicitly clarified in the methods and limitations sections, emphasizing the rationale and impact on generalizability.
Comment 2: All definitions have been included which is great, though some fine adjustments in the interpretation of the data need to be clarified. Limitations are well described and as outlined by Reviewer 2, the significant heterogeneity of the studies need to be addressed.
Response: Thank you for pointing this out. We acknowledge the significant heterogeneity in the included studies, as highlighted by Reviewer 2. To address this, we expanded our limitations section. We now explicitly discuss how differences in surgical techniques (open vs. laparoscopic), anesthetic approaches, and patient populations (gestational age, comorbidities) contribute to variability in outcomes. Additionally, we propose a framework for standardizing future research to reduce such heterogeneity and improve the comparability of findings.
We hope that this revision addressed all the respected editor’s and reviewers' concerns and improves the quality of the manuscript. Please let us know if any further clarifications are required.
Thank you for your time and consideration.
Sincerely,
On behalf of the authors,
Corresponding Author
Dr. Arwa Alsharif
14-January-2025
